Validation of picogram- and femtogram-input DNA libraries for microscale metagenomics

Rinke Christian 1 christian.rinke@gmail.com
Low Serene 1
http://orcid.org/0000-0003-0670-7480 Woodcroft Ben J. 1
Raina Jean-Baptiste 2
Skarshewski Adam 1
Le Xuyen H. 1
Butler Margaret K. 1
Stocker Roman 3
Seymour Justin 2
Tyson Gene W. 1 4
Hugenholtz Philip 1 5 phugenholtz@gmail.com
1 Australian Centre for Ecogenomics/School of Chemistry and Molecular Biosciences, University of Queensland , Brisbane, QLD , Australia
2 Climate Change Cluster, University of Technology Sydney , Sydney, New South Wales , Australia
3 Department of Civil, Environmental and Geomatic Engineering, ETH Zurich , Zurich , Switzerland
4 Advanced Water Management Centre, University of Queensland , Brisbane, QLD , Australia
5 Institute for Molecular Bioscience, University of Queensland , Brisbane, QLD , Australia
Smidt Hauke
Electronic publication date: 2016 Sep 22
Publication date: 2016
Volume: 4
Electronic Location ID: e2486
Received 2016 Jun 14; Accepted 2016 Aug 24
Copyright: © 2016 Rinke et al.
Copyright year: 2016
Copyright holder: Rinke et al.
License: This is an open access article distributed under the terms of the Creative Commons Attribution License, which permits unrestricted use, distribution, reproduction and adaptation in any medium and for any purpose provided that it is properly attributed. For attribution, the original author(s), title, publication source (PeerJ) and either DOI or URL of the article must be cited.
License URL: https://creativecommons.org/licenses/by/4.0/

Keywords: Nextera XT, 100 fg, Low input DNA library, Picogram, Reagent contamination, Low biomass, Low volume, Microscale metagenomics, Marine microheterogeneity, Illumina

Funding: Gordon and Betty Moore Foundation GBMF3801 Australian Research Council Laureate Fellowship FL150100038 Genomic Science Program of the United States Department of Energy Office of Biological and Environmental Research grant DE-SC0004632 Australian Research Council Discovery Early Career Research Award #DE160100248 This work was primarily funded by the Gordon and Betty Moore Foundation (Grant ID: GBMF3801). PH was also supported by an Australian Research Council Laureate Fellowship (FL150100038), and BJW and GWT by the Genomic Science Program of the United States Department of Energy Office of Biological and Environmental Research, grant DE-SC0004632. BJW was also supported by Australian Research Council Discovery Early Career Research Award #DE160100248. The funders had no role in study design, data collection and analysis, decision to publish, or preparation of the manuscript.

==============================
High-throughput sequencing libraries are typically limited by the requirement for nanograms to micrograms of input DNA. This bottleneck impedes the microscale analysis of ecosystems and the exploration of low biomass samples. Current methods for amplifying environmental DNA to bypass this bottleneck introduce considerable bias into metagenomic profiles. Here we describe and validate a simple modification of the Illumina Nextera XT DNA library preparation kit which allows creation of shotgun libraries from sub-nanogram amounts of input DNA. Community composition was reproducible down to 100 fg of input DNA based on analysis of a mock community comprising 54 phylogenetically diverse Bacteria and Archaea. The main technical issues with the low input libraries were a greater potential for contamination, limited DNA complexity which has a direct effect on assembly and binning, and an associated higher percentage of read duplicates. We recommend a lower limit of 1 pg (∼100–1,000 microbial cells) to ensure community composition fidelity, and the inclusion of negative controls to identify reagent-specific contaminants. Applying the approach to marine surface water, pronounced differences were observed between bacterial community profiles of microliter volume samples, which we attribute to biological variation. This result is consistent with expected microscale patchiness in marine communities. We thus envision that our benchmarked, slightly modified low input DNA protocol will be beneficial for microscale and low biomass metagenomics.

Introduction

Over the last decade, advances in high-throughput sequencing technologies have accelerated the exploration of the uncultured microbial majority (Rappe & Giovannoni, 2003). The direct sequencing of environmental samples, termed metagenomics, has revolutionized microbial ecology by providing new insights into the diversity, dynamics and metabolic potential of microorganisms. A remaining limitation of conventional metagenomic library construction is the requirement for relatively large sample amounts, e.g. grams of soil or liters of seawater which comprise millions of microbial cells. However, samples of this size aggregate microbial population heterogeneity and metabolic processes occurring at the microscale.

Within natural habitats, specific physiological niches occupied by microorganisms often occur within discrete microenvironments that are several orders of magnitude smaller than typical sample sizes. For example, in the pelagic ocean dissolved and particulate organic matter is often localized within hotspots, ranging in size from tens to hundreds of micrometers (Azam, 1998). These hotspots include marine snow particles, cell lysis and excretions by larger organisms such as phytoplankton exudates, which result in microscale chemical gradients that chemotactic bacteria can exploit (Azam & Malfatti, 2007; Stocker, 2012). Specific populations and their associated biogeochemical activities can be restricted to these localized microniches (Paerl & Pinckney, 1996). Therefore, understanding processes occurring at the microscale (μg, μl) is important if we are ever to fully understand ecosystem functionality. Beyond the need for increased ecological resolution, there is a demand for creating metagenomes from small amounts of starting DNA to explore habitats with extremely low biomass such as subseafloor sediments (Kallmeyer et al., 2012), clean-room facilities (Vaishampayan et al., 2013), human skin samples (Probst, Auerbach & Moissl-Eichinger, 2013), and ocean virus samples (Duhaime et al., 2012).

Template preparation for high throughput sequencing platforms traditionally follows a common workflow, independent of the downstream sequencing chemistry. First the input DNA is sheared to fragments of the desired size by random fragmentation and subsequently platform-specific sequencing adapters are added to the flanking ends in order to attach the library to a solid surface (e.g. flow cell, tagged glass slide, bead) via a complementary sequence. Typically, the input DNA is sheared by sonication, followed by multiple rounds of enzymatic modification to repair the DNA fragments to have blunt ends or a tails and to add the sequencing adapters. This method is labor intensive and requires several tens of nanograms to micrograms of input DNA, making it challenging to prepare sequencing libraries from low yield DNA samples. Linker-amplification comprising ultrasonic shearing, linker ligation and PCR amplification is another approach that has been applied to create low input DNA shotgun libraries from ≤ 1 ng starting DNA (Duhaime et al., 2012; Solonenko et al., 2013). However, this method is time consuming, technically demanding, and known to introduce up to 1.5-fold GC content amplification bias (Duhaime et al., 2012). Multiple displacement amplification (MDA) with phi29 polymerase can increase DNA amounts by nine orders of magnitude allowing femtogram range DNAs to be used for library preparation (Raghunathan et al., 2005). While MDA is successfully applied to obtain single-cell genomes (Clingenpeel et al., 2015), the approach has been shown to significantly skew microbial community profiles (Yilmaz, Allgaier & Hugenholtz, 2010; Probst et al., 2015).

The recent development of the Nextera™ technology substantially speeds up Illumina library creation and reduces input DNA requirements down to 1 ng (Nextera-XT). In this approach DNA is simultaneously fragmented and tagged (“tagmentation”) using in vitro transposition (Syed, Grunenwald & Caruccio, 2009; Caruccio, 2011). The resulting tagged fragments undergo a 12-cycle PCR reaction to add sequencing adaptors and sample-specific barcodes, which facilitate sample multiplexing. A number of attempts have been made to push the limits of Nextera library creation into the sub-nanogram range including the creation of unvalidated libraries from 10 pg of human DNA (Adey et al., 2010) and validated libraries using as little as 20 pg of E. coli and mouse genomic DNA (Parkinson et al., 2012). The latter study found that this technique provided deep coverage of the E. coli K-12 genome, but also increased the proportion of duplicate reads and resulted in over-representation of low-GC regions. Most recently, the fidelity of picogram-level libraries was assessed using a simple mock microbial community, which found minimal impact of input DNA (down to 1 pg) on community composition estimates using the Nextera-XT kit (Bowers et al., 2015). Here, we extend this approach using a more complex mock community and environmental samples down to the femtogram input DNA range. We find that read-mapping estimates of community composition fidelity are reproducible down to 100 fg and infer that variance in community structure between replicate 10 μl marine samples appears to be primarily due to microscale biological differences.

Materials and Methods

Mock community construction

Genomic DNA extracted from 40 bacterial and 14 archaeal taxa for which reference genomes are available (54 isolate genomes), were combined to create a mock community (Table S1). Purified genomic DNAs from 49 of the 54 mock community members were obtained from collaborators. These DNAs were assessed via gel electrophoresis and quantified with qPCR (Shakya et al., 2013). The DNAs from the remaining five cultures were harvested and purified in our laboratory (see ‘Assembly of mock community genomes’ below). The amount of DNA and the genome size of each isolate were used to calculate their expected relative abundance in the community. The organisms for which only low amounts of gDNA were available were added in lower abundances to the final mix (Table S1). The final DNA concentration of the mock community was 23.1 ng/μl, which was diluted appropriately for low input library construction.

Marine sampling

Marine surface seawater samples were obtained from Blackwattle Bay in Sydney Harbor (33°52′S, 151°11′E). Seawater was collected in a 10 L sampling container and low volume samples (1 ml, 100 and 10 μl) were pipetted individually from the 10 L sample and snap frozen directly at the sampling site. For the marine standard operating procedure (SOP), triplicate 10 L samples were collected and transported to the laboratory (∼30 min travel time). Upon arrival, the samples were pre-filtered through a 10 μm filter (Millipore, Billerica, MA, USA) to remove large particles and subsequently filtered through 0.2 μm Sterivex filters (Millipore, Billerica, MA, USA). All samples were kept at −80 °C until further processing. The entire sample preparation and analysis workflow is shown in Fig. S1.

DNA extraction from seawater samples

DNA extraction from seawater samples was performed using the UltraClean® Tissue & Cells DNA Isolation Kit following manufacturer’s instructions. Minor modifications were made to optimize the DNA extraction for marine samples and the low volume samples (102–106 cells/ml), and for consistency were also applied to the filtered marine SOP samples. Briefly, 1 ml (instead of 700 μl) Solution TD1 was added directly into the low volume seawater samples (1 ml, 100 and 10 μl; all in 1.5 ml tubes) or directly into the Sterivex filter (10 L filtered SOP). Using a Vortex-Genie®, samples were lyzed and homogenized by vortexing the tubes and filters respectively at maximum speed for 1 min, without adding the recommended beads. Finally, 20 μl (instead of 50 μl) of elution buffer was added and incubated at room temperature for 5 min before centrifugation. For the low input libraries, 1/4 of the DNA extraction volume (5 μl) was used for library creation. Thereby, the amount of input DNA for library preparation was quantified, using a Qubit-fluorometer (Invitrogen), for the SOP and the 1 ml libraries, and was estimated for the 100 and 10 μl samples based on the 1 ml sample measurements. The number of cells in the low volume samples was calculated based on an average DNA content of 1–10 fg per cell.

DNA library preparation

Libraries were prepared using the Nextera XT DNA Sample Preparation Kit (Illumina Inc., San Diego, CA, USA). The standard protocol of the manufacturer was modified to optimize library preparation from DNA input concentrations of less than 1 ng (0.2 ng/μl). The amplicon tagment mix (ATM), which includes the enzyme used for tagmentation, was diluted one in 10 in nuclease free water. For each sample, a 20 μl tagmentation reaction contained 10 μl TD buffer, 5 μl of input DNA and 5 μl of the diluted ATM. Tagmentation reactions were incubated on a thermal cycler at 55° for 5 min. Subsequently, tagmented DNA was amplified via a limited-cycle PCR whereby the number of amplification cycles was increased from 12 to 20 cycles to ensure sufficient library quantity for the downstream sequencing reaction. Amplified libraries were purified with 1.6× Ampure XP beads and eluted in 20 μl of re-suspension buffer. The quality of the purified libraries was assessed using the High Sensitivity DNA kit on the Agilent 2100 Bioanalyzer. Successful libraries were quantified through qPCR using the KAPA Library Quantification Kits, according to manufacturer’s instructions, prior to pooling and sequencing. The creation of each low input library was performed in triplicate, together with a negative control containing no input DNA.

DNA sequencing

All libraries were sequenced with an Illumina NextSeq500 platform 2× with 150 bp High Output v.1 run chemistry. The replicate SOP, 100, 10 and 1 pg, 100 fg, negative control, and marine sample libraries were pooled on an indexed shared sequencing run, resulting in 1/37 of a run or ∼3.2 Gb per sample. The adapter trimmed fastq read files were deposited on the Microscale Ocean webpage (http://microscaleocean.org/data/category/9-low-input-dna-libraries-peerj).

Genome reference database

A genome reference database was created by concatenating the fasta files of the 54 mock community member genomes (Table S1), the M. aerolatum contaminant genome, the human genome (release GRCh37), and the phiX 174 genome.

Read mapping based mock community profiles

Adapter trimmed sequences were aligned against the genome reference database using BWA MEM 0.7.12 (Li, 2013) through BamM (http://ecogenomics.github.io/BamM/). To improve stringency the seed length was increased to 25 bases in BWA MEM mode (–extras “mem:-k 25”). The resulting bam files were evaluated with samtools (Li et al., 2009), using samtools view (http://www.htslib.org/) and a custom script counting the mapped reads per reference genome.

Insert size

The BamM generated bam files (see read mapping above) were randomly subsampled to one million aligned read pairs and the CIGAR string column nine (TLEN, observed template length) was extracted, using samtools view and the GNU coreutils command-line programs awk and shuf. Trimmed mean (trim = 0.01) and trimmed standard deviation (trim = 0.01) were calculated with the Rstudio package (https://www.rstudio.com/). The applied definition of the term “insert size” used throughout this manuscript is the number of bases from the leftmost mapped base in the first read to the rightmost mapped base in the second read.

Read %GC content

Raw FASTQ-format forward reads were converted to FASTA format and the %GC content calculated with a custom perl script for each library replicate. The first 10,000 reads per replicate were used to calculate the average %GC content.

Read duplicates

A custom python script (checkunique_v7.py) was used to estimate the percentage of read duplicates using raw reads as input. The script loads a forward and a reverse read FASTQ file, randomly selects a given number of read pairs, and concatenates the first 30 bp from the forward and reverse reads into a 60 bp sequence. The 60 bp sequences from different read pairs are then compared and the number of unique pairs is recorded. Read duplicates are defined as total counted read pairs minus unique pairs. The script takes increments as optional arguments, and performs subsampling (e.g. in 100,000 read increments) and subsequent counting of unique (duplicate) read pairs per subsample, which allows plotting of a read duplicate rarefaction curve. The cutoff of 30 bases per read and 100% match was chosen after initial trials showing that this cutoff is comparable to read duplicate levels estimated by read mapping to reference genomes of the mock data set (data not shown). For reference-based read mapping, raw reads were subsampled with seqtk (https://github.com/lh3/seqtk) mapped against the reference genome database with BamM and duplicate read pairs were removed with samtools rmdup (Li et al., 2009), which defines a duplicate as a read with the exact same start and stop position as an already mapped read, and compared to the same file before duplicate removal using samtools flagstat.

Taxonomic profiles

The 16S rRNA gene-based taxonomic profiles of the mock community and seawater samples were generated with GraftM (http://geronimp.github.io/graftM) using the 16S rRNA package (4.06.bleeding_edge_2014_09_17_greengenes_97_otus.gpkg). The pipeline was designed to identify reads encoding 16S rRNA genes based on HMMs and to assign taxonomic classifications by comparing against a reference taxonomy. A detailed feature description, user manual, and example runs are available on the GitHub wiki (https://github.com/geronimp/graftM/wiki). For the heat map, the GraftM output was manually curated, whereby mitochondrial and chloroplast sequences were removed. Taxon counts were trimmed (max > 20), and analyzed with DESeq2, a method for differential analysis of count data using shrinkage estimation for dispersions and fold changes (Love, Huber & Anders, 2014), in the software environment R (www.r-project.org). The data were log transformed (rlog), and displayed as a heat-map (pheatmap).

Functional profiles

Reads were searched against uniref100 (Suzek et al., 2007) (accessed 20151020) using DIAMOND v0.7.12 (Buchfink, Xie & Huson, 2015) with the BLASTX option. The top hit of each read (if above 1e-3) was mapped to KEGG Orthology (KO) IDs using the Uniprot ID mapping files. Hits to each KO were summed to produce a count table. Correlations and significance tests were performed with R (www.r-project.org) after applying and a cut-off > 500.

Assembly and binning

Reads were adapter trimmed and subsampled as follows. For the replicate assemblies, forward and reverse reads were subsampled to five million reads each. For the combined assemblies, reads from each replicate of a library were combined and then subsampled to 25 million reads. Assemblies were performed with the CLC Genomics Workbench 8.0.2 (http://www.clcbio.com) using default settings and a 1 kb minimum contig size. Binning of population genomes was performed with MetaBAT using default settings (–sensitive) as described previously (Kang et al., 2015), and the resulting population genome bins were evaluated and screened with CheckM (Parks et al., 2014).

Sequence logo generation

The mock community library forward reads were subsampled to 1,000 reads using seqtk (https://github.com/lh3/seqtk). Reads were trimmed to the first 10 bases with a custom Perl script (trimFasta.pl), and the reads were submitted to weblogo (Crooks et al., 2004) to generate sequence logos (Schneider & Stephens, 1990).

Statistical analysis

The software packages MYSTAT and SYSTAT (http://www.systat.com) and R (www.r-project.org) were used for all statistical analyses. Datasets were analyzed by ANNOVA (parametric) and Tukey’s Significance Test, the non-parametric Kruskal-Wallis One-way Analysis of Variance, or the Bonferroni probabilities (p-value) for correlations. Results from the 16S rRNA gene based community profiles and the functional profiles were used to calculate the mean coefficient of variation, which is defined as the ratio of the standard deviation to the mean.

Assembly of mock community genomes

The five draft genomes were obtained by harvesting 10 ml of culture medium and extracting the DNA using the PowerSoil® DNA Isolation Kit (MO BIO Laboratories). Genome sequencing was performed on the Illumina NextSeq 500 platform using the Nextera library protocol. Raw sequencing reads were adapter clipped and quality trimmed with trimmomatic (http://www.usadellab.org/cms/?page=trimmomatic) 0.32 using the parameters “LEADING:3 TRAILING:3 SLIDINGWINDOW:4:15 CROP:10000 HEADCROP:0 MINLEN:50” with Nextera adapter sequences. BBMerge (version BBMAP: bbmap_34.94; https://sourceforge.net/projects/bbmap) was used to merge overlapping pairs of reads using default parameters. Quality controlled paired reads were assembled with CLC Genomics Cell assembler v4.4 using an estimated insert size of 30–500 bp. Quality controlled paired reads were mapped to the assembled contigs using BamM v1.5.0 (http://ecogenomics.github.io/BamM/), BWA 0.7.12 (Li, 2013) and samtools (http://www.htslib.org/) 0.1.19. The coverage of each contig was determined with BamM ‘parse’ and the average coverage weighted by contig length estimated. To reduce the deleterious effects of excess sequencing depth, raw reads were subsampled to provide an estimated 100× coverage using seqtk ‘sample’ (https://github.com/lh3/seqtk), quality controlled with trimmomatic/BBMerge and reassembled. Contigs less than 2 kb were removed and genome quality assessed with CheckM 1.0.3 (Parks et al., 2014) using ‘lineage_wf.’ All genomes were assessed as being > 95% complete and < 1% contaminated. The resulting five population genome sequences were deposited in NCBI-BioProject under the BioProject ID PRJNA324744.

Results

Low input library and sequence data quality

The Nextera-XT SOP is suitable for library preparation of 1 ng input DNA, which is equivalent to 105–106 microbial cells, assuming 1–10 fg per cell (Button & Robertson, 2001). We began by testing the Nextera-XT kit with 1 pg input DNA from E.coli and a mock microbial community, equivalent to 102–103 cells, a relevant range for localized microenvironments found within natural marine systems. The mock community comprised 54 bacterial and archaeal isolates with sequenced genomes and GC contents ranging from 28 to 70%, which were pooled at relative abundances ranging from 0.04 to 25% (Table S1). For 1 pg input DNA, it was necessary to increase the limited cycle PCR from 12 to 20 cycles to ensure sufficient tagmentation amplicon product for library preparation (Bowers et al., 2015) and sequencing (Fig. S2). The ATM used in the Nextera-XT kit includes a transposase that fragments and tags the input DNA during the tagmentation process, and we hypothesized that over-fragmentation of low input DNA could be avoided by diluting the enzyme and/or decreasing the tagmentation reaction time. We found a direct correlation between ATM dilution and insert size for both the pure and mixed templates, ranging from shorter than desired fragments (< 200 bp) in the 1:5 dilutions to fragment sizes equal to the 1 ng SOP in the 1:50 dilutions (∼300 bp; Figs. 1 and S3). Surprisingly, the 1 pg undiluted ATM controls produced insert sizes similar to the 1:10 dilutions (∼240 bp) in contrast to the anticipated over-fragmentation observed in the 1:5 dilutions (see Discussion).

Figure 1 Dilution series evaluation for low input DNA libraries.

Dilutions down to 1:50 of amplicon tagment mix (ATM) for low input DNA libraries of 1 pg DNA are shown in comparison to the 1ng SOP. The trimmed mean insert size (length of DNA fragments in bases without adaptors, determined via read mapping) is plotted against the relative number of read duplicates. Libraries were created with 1 pg of (A) E. coli DNA, (B) Mock community DNA. Reads were subsampled to five million read pairs. Note that E. coli was not sequenced with the 1 ng SOP.

Next we investigated the percentage of read duplicates in each library, a known artifact of the limited cycle PCR step (Kozarewa et al., 2009), which may compromise de novo assembly (Xu et al., 2012). In general, the percentage of read duplicates was much higher in the 1 pg libraries than the 1 ng SOP libraries; ∼50 vs ∼2%, respectively at a sampling depth of five million read pairs (Fig. 1). The percentage of read duplicates increased with increasing ATM dilution (Fig. 1), so to achieve an insert size of > 200 bp (recommended by Illumina for Nextera-XT libraries) while minimizing read duplicates, we proceeded with the 1:10 dilution. This ATM dilution gave similar results to the undiluted samples in terms of insert size and read duplicates (Fig. 1), but provides the potential to create multiple libraries from the same ATM starting volume.

We next tested a range of low input DNA concentrations (100 pg to 100 fg) using the mock community and our modified protocol (20 cycles, 1:10 ATM dilution) prepared in triplicate, and evaluated by comparison to 1 ng SOP libraries. Library creation was reproducibly successful down to 1 pg as assessed by Bioanalyzer and qPCR profiles (Fig. S4). Two of the three initial 100 fg libraries were successful, but only slightly above the Bioanalyzer detection limit and showed a lag in qPCR amplifications of about two cycles compared to the 100 pg library (Fig. S4). Therefore, an additional three 100 fg libraries were created, all of which progressed to sequencing. To assess possible contamination we included two types of negative controls by substituting ddH2O for input DNA in the DNA extraction and library construction steps. We selected the UltraClean DNA extraction kit for the DNA extraction control as this was the kit used to extract marine samples (see Application to environmental samples below). Negative controls showed DNA detectable via the Bioanalyzer and/or qPCR assays and were therefore sequenced (Fig. S5). The sequencing runs were successful for all low input libraries (with the exception of the one 100 fg library), producing between 12 and 22 million adaptor-trimmed reads, which slightly less than the 1 ng SOP libraries (30.5 mil; SD ± 6.9 mil), with a trend towards decreasing sequence yield with decreasing library input DNA (Fig. 2). The no input DNA negative controls resulted in a higher yield for the DNA extraction plus library kit control (13.8 mil; SD ± 3.8 mil) compared to low read numbers for the library only kit control (5.7 mil; SD ± 3.7 mil, Fig. 2).

Figure 2 Yield and quality assessment of low input libraries.

The bar graph shows the absolute number of reads for all replicates of the 1 ng SOP, the low input libraries (100pg, 10pg, 1pg, 100fg) and the negative controls (grey background). Negative controls are comprised of the library prep kit control (NegLib) and the DNA extraction kit + library prep kit control (NegExt), see Methods for details. Reads are colour coded based on the reference they aligned to, including the bacterial and archaeal mock community (green) and the human genome (blue). The remaining reads are shown as unmapped (orange) or mapped against the contaminant Methylobaterium aerolatum (red). The calculated cell number range (∼no. cells) is based on the amount of input DNA and an estimated 1–10 fg DNA per microbial cell. The sequence yield is provided as million reads (read yield). The average insert size (insert size), the average percent GC content (%GC), and the average number of read duplicates (% duplicates) was calculated as a mean of all replicates. The bar above the figure indicates when the standard protocol (SOP) or our modified protocol was used for library creation. The bar below the figure provides the average expected reads per sample, based on a NextSeq500 2× 150 bp High Output v. 1 run with 1/37 sequence allocation per library. Sample replicate numbers are given in parenthesis.

The average insert size of the low input mock community libraries, as estimated by read mapping to the reference genomes, was significantly smaller (p < 0.01) than the SOP library, with 307 ± 115 bp observed in the SOP libraries vs 204–257 bp in the low input libraries (Fig. 2; Table S2). Although the average read GC content was not significantly different (p < 0.05) between the 1 ng SOP and low input libraries, an appreciable drop was recorded for the 100 fg libraries (Figs. 2 and S6). Read duplicates increased with decreasing input DNA from an average of 1.2% for the 1 ng SOP up to 78.8% for 100 fg, at a sampling depth of five million read pairs (Figs. 2 and S7A). This is consistent with the higher levels of read duplicates observed in the initial 1 pg E.coli and mock community libraries (Fig. 1). We noted that read duplicates are library-specific as combining reads from the 1 pg library replicates sampled to the same depth reduced the proportion of duplicates and improved assembly statistics (Figs. S7B and S8). This library specificity suggests that successfully tagmented DNA is a random subset of input DNA, which in the case of low input samples increases the proportion of duplicates by further limiting the template for limited cycle PCR. We also explored the relationship between percentage read duplicates and the limited cycle PCR step, whereby we raised the number of cycles from 12 to 20. We predicted that lower cycles should produce a lower proportion of read duplicates. Additional replicated low input libraries were prepared using 12, 14, 16, and 18 cycles, according to Bioanalyzer detectability thresholds, for a given amount of input DNA (Fig. S9). In contrast to expectations, the percentage of read duplicates did not change appreciably as a function of PCR cycle for a given DNA input concentration (Fig. S10), suggesting that they are mostly created within the first 12 cycles.

Sequencing contaminants

To determine the identity of possible sequencing contaminants, we assembled the reads from all negative control libraries. A substantially complete Methylobacterium genome (87.9% according to CheckM; Parks et al. (2014) was assembled which had 100% identity over 758 bp to the 16S rRNA gene of M. aerolatum strain 92a-8 (Weon et al., 2008). Members of the genus Methylobacterium are recognized reagent contaminants (Salter et al., 2014). This genome was used to aid in identifying contaminant reads in the mock community libraries (Fig. 2). We aligned the sequence reads of the mock community and negative controls against the reference database containing all 54 microbial genomes, the M. aerolatum contaminant genome, the human genome to detect possible operator contamination and the phiX 174 genome, which is used as an internal run quality control during Illumina sequencing. For the 1 ng SOP and 100, 10 and 1 pg low input libraries, 95.6–97.7% of reads mapped to the microbial mock community and small percentages mapped to the human reference (0.1–1.9%) or were unmapped (2.1–3.9%; Fig. 2). We found more variable results for the five 100 fg libraries: two libraries produced similar results to the higher input libraries (rep1 & rep2 in Fig. 2) and the other three had high human contamination (30.4–45.0% of reads; rep3, rep4 & rep5 in Fig. 2). Reads from the negative control libraries were mostly human contamination (11.0–60.5%), M. aerolatum (0.4–65.5%), or unmapped (16.0–48.6%). The three UltraClean kit negative control libraries accounted for 98.5% of all reads mapping to M. aerolatum suggesting that this DNA extraction kit is the primary source of this contaminant. A small proportion of the negative control reads (0.3–0.9%) mapped to the mock community, which we attribute to cross sample contamination with mock community libraries due to false index pairings of the multiplex sequencing runs (Kircher, Sawyer & Meyer, 2012). A closer examination of the unmapped negative control reads (16.0–48.6%) revealed the presence of mostly Firmicutes (Bacilli and Clostridia), but also Proteobacteria, Actinobacteria and Elusimicrobia (Fig. S11). We did not detect phiX 174 contamination (data not shown).

Fidelity of community composition

We evaluated community composition fidelity by comparing the relative abundance of the 54 mock community members between the 1 ng SOP library and the low input libraries, based on read mapping to the mock community genome database. Community composition of low-input libraries averaged across replicates (excluding library 100 fg_S12) was strongly (R2 ≥ 0.94) and significantly (p < 0.001) correlated to standard input libraries (Fig. 3; Table S3A), indicating that reducing input DNA resulted in minimal representational bias. This significant correlation was upheld even when the five most abundant community members were excluded, although the 100 fg libraries began to show slightly higher variance (Fig. 3; Table S3B). Since reference genome sets are usually unavailable for environmental samples, we also assessed community composition using 16S rRNA-based taxonomic profiling and functional profile analysis using the KEGG database. The 16S rRNA-based taxonomic profiles matched the genome-based read mapping analysis with all libraries being strongly (R2 ≥ 0.98) and significantly (p < 0.001) correlated to each other (Fig. S12). Functional profiles between SOP and low input libraries down to 1 pg were also strongly (R2 ≥ 0.99) and significantly (p < 0.001) correlated, with lower but still significant correlations to the 100 fg libraries (R2 ≥ 0.91; p < 0.01; Fig. S13). We therefore suggest that community composition can be reliably assessed using low input libraries down to 100 fg despite higher proportions of read duplicates (Fig. S7A) provided contaminants are accounted for by including negative controls. To determine if the consistently reduced correlation between the 100 fg and SOP libraries was a systematic effect of the observed difference in average read %GC (Fig. 2), we investigated the relative abundance of community members with high, median and low genomic GC content. No significant differences (p > 0.45) were observed between the relative abundances of reads aligning to high, medium or low GC genomes among all input libraries compared to the SOP (Table S4). This suggests that there is no substantial bias against high GC organisms among our mock community members with decreasing input DNA, although there was a slight trend in this direction (Fig. S14). An analysis of the regions immediately flanking the transposase insertion sites (first 10 bases of each read) indicated a slight preference for insertion into AT-richer regions relative to the mean GC content (Fig. S15). However, this preference was consistent between the different input DNA concentrations and should have had no net effect on average read %GC content.

Figure 3 Mock community profile comparisons.

Correlation between the 1 ng SOP libraries (x-axes) and the low input DNA libraries (100, 10 and 1 pg, 100 fg; y-axes). Shown is the mean relative abundance of the 54 mock community members, based on reads aligned to the respective reference genomes.Inserts: show a subset of the relative abundances excluding the five most dominant organisms of the mock community. The mean standard deviation for each library is provided as error bars. The 100 fg libraries include four replicates (1, 2, 4, 5) out of five, omitting replicate 3 which was highly contaminated.

Assembly and binning

We normalized each replicate dataset to five million read pairs and assembled them using CLC Genomics workbench (see Materials and Methods). Assembly quality metrics–maximum contig length, total assembly size and number of contigs–were found to deteriorate with decreasing input DNA (Fig. 4). The increasing percentage of read duplicates with decreasing input DNA (Fig. S7A) may be primarily responsible for this observed drop in assembly statistics. However, when duplicates were removed, assembly statistics did not improve relative to the corresponding assemblies using all reads (Fig. 4) suggesting that read duplicates per se did not hamper the assembly. Instead, we noted a strong correlation between decreasing percentage of unique reads and loss of assembly performance for the mock community datasets (Fig. S16). Despite this loss in performance, datasets down to 1 pg still produced thousands of contigs > 1 kb (up to a maximum of 260 kb) from five million read pairs suitable for population genome binning. To assess maximum binning potential for the mock community datasets, we combined and co-assembled replicates (50 million reads) except for the 100 fg libraries which did not meet this sequence read threshold. A drop in assembly performance was also noted with decreasing input DNA for the larger datasets and the removal of read duplicates had no effect on assembly statistics (Fig. S17). However, nine moderately complete (> 50%) to near (> 90%) complete genomes with low contamination (< 10%) were still recovered from the 1 pg library assembly, compared to 24 for the SOP library (Table 1).

Figure 4 Mock community assembly statistics.

(A) Maximum contig size, (B) total assembly size, (C) number of contigs, and (D) N50 of the SOP and low input mock community libraries. Read files were subsample to five million read pairs. Gray bars show assemblies of all reads, red bars show assemblies after read duplicates were removed. Only contigs ≥ 1 kb were included in the analysis. All values are given as mean and standard deviation.

Table 1 Mock community population genome bins.

Read were subsample to 50 million reads, and only contigs ≥ 1 kb were used for population genome binning. Completeness and contamination were estimated based on marker genes, see Material and Methods.

	Assembly		Completeness	Contamination	
Size	max contig	bins	mean	max	min	mean	max	min	
mock_SOP_1ng	148,892,162	952,162	24	83.116	99.63	54.62	0.898	6.62	0	
mock_100pg	132,682,229	866,524	17	87.504	99.68	53.07	1.136	2.64	0	
mock_10pg	115,499,380	852,918	13	83.87	99.45	54.41	1.812	6.9	0	
mock_1pg	54,332,814	522,839	9	85.994	98.86	57.55	0.978	2.03	0	

Application to environmental samples

Based on the observation that the mock community profiles were consistent between the SOP and low-input DNA library protocols, we applied our approach to marine surface water samples (Sydney Harbor, Australia). In order to obtain sufficient DNA for the Nextera XT SOP, we filtered 10 L of surface water obtaining > 100 ng bulk DNA. Small unfiltered volumes, obtained from a 10 L surface water sample, comprising 1 ml, 100 and 10 μl, were used to create low-input DNA libraries. DNA was extracted from replicated samples using the UltraClean kit, applying a minor modification to accommodate marine samples and the smaller unfiltered volumes (see Materials and Methods). The bulk DNA from the 10 L filtered sample was used for the SOP libraries and diluted to create low input DNA libraries equivalent to the low sample volume libraries. Library creation was successful for all samples and yielded between 14.8–30.2 million adaptor-trimmed reads for the low input samples compared to 24.6 ± 1.2 million reads for the equimolar pooled 10 L filtered SOP samples (Fig. 5). To detect possible contamination, we aligned the reads against the reference genome database (see Materials and Methods). No phiX contamination was found, but varying degrees of human and M. aerolatum contamination were detected, with the general trend of increasing contamination with decreasing input DNA (Fig. 5) as seen in the mock community samples (Fig. 2). Substantial contamination was noted in four of the low volume libraries (≤ 27.3% reads) highlighting the importance of i) running negative control libraries to identify contaminants, some of which may be kit or reagent specific, and ii) screening low-input DNA shotgun datasets for identified and known contaminants.

Figure 5 Yield and quality assessment of marine samples.

Reads are color coded based on the reference they aligned to, including the known contaminant Methylobacterium aerolatum (red) and the human genome (blue). The remaining reads are shown as unmapped (orange). The amount of DNA extracted with the modified extraction protocol is given as total DNA in 20 µl elution buffer (DNA extract). Number of cells (∼no. cells) was calculated based on an average DNA content of 1–10 fg per cell. The amount of input DNA for library preparation was measured for the SOP and the 1 ml libraries, and was estimated for the 100 and 10 µl samples based on the 1 ml sample measurements. The bar above the figure indicates when the standard protocol (SOP) or our modified protocol was used to create the libraries. All libraries were sequenced at an allocation of 1/37 of an Illumina NextSeq500 2×150 bp High Output v. 1 run. Sample replicate numbers are given in parenthesis.

After removing contaminant reads, we used 16S rRNA-based taxonomic and KO-based functional profile analyses to evaluate community reproducibility between the different libraries in the absence of reference genomes for these samples. The inferred taxonomic profiles were consistent with those expected within surface marine water samples including a dominance of populations belonging to the Pelagibacteraceae, Flavobacteriaceae, and Synechococcaceae families (Figs. 6 and S18). Profiles averaged across replicates of both the filter dilution and low volume marine communities were strongly correlated to the SOP libraries down to 5 pg and 10 μl, respectively (Figs. 7, S19 and S20). Replicate profiles were also highly correlated with the exception of the 10 μl libraries (Figs. 8 and S18). This exceeded the expected technical variation observed in the corresponding 5 pg filter dilutions according to comparisons based on the mean coefficient of variation (Fig. 9).

Figure 6 Abundance profiles of the marine microbial samples.

Bacterial taxonomy was assigned based on 16S rRNA gene sequence detection of shotgun sequencing reads (graftM; see Methods). The normalized abundance is shown after square root transformation for all OTUs above the abundance threshold, resulting in a normalized read count (NR) from 0 to 800. The taxonomic assignment is provided down to the family level if available, otherwise the best available taxonomic rank is given.

Figure 7 Marine communities profile correlations.

Correlation coefficients are shown for the marine 10 L SOP, the 10 L filtered dilution, and the low input DNA libraries. The panels show the 16S rRNA gene based taxonomic profile correlations, and the KO-based functional profile correlations. The Pearson correlation coefficient is colour coded from zero (white) to one (dark blue).

Figure 8 Profile analyses of marine sample replication.

Replicate correlation plots of (A) 16S rRNA gene based taxonomic profiles and (B) functional KO based profiles. Samples with comparable DNA input amounts are connected via a grey box.

Figure 9 Mean coefficient of variation for taxonomic marine community profiles.

The mean coefficient of variation is applied to compare the 10 L dilutions (SOP 1 ng, 50 and 5 pg) against the low volume (1 ml, 100 and 10 ul) samples using 16S based taxonomic profiles. The X-axis shows the different amounts of input DNA and volumes for the low volume samples (upper row) and the 10 L filtration (SOP, and dilutions; lower row).

Assembly and binning were also assessed for the marine samples. We repeated the approach used for the mock community and found that the assembly and binning of the 10 L filtered dilution libraries (5 and 50 pg) produced similar results to the 10 L filtered SOP (Fig. S21; Table S5). However, the low volume libraries representing between ∼3 pg (10 μl) and ∼300 pg (1 ml) input DNA had poorer assembly metrics (Fig. S21) and did not produce any population genomes with > 50% completeness as compared to the 10 L filtered datasets which produced five such genomes (Table S5).

Discussion

The typical requirement of microgram quantities of DNA for metagenome library creation often necessitates the use of bulk samples comprising millions to billions of individual microbial cells in macroscale quantities (ml, g). Low input DNA library creation protocols provide the opportunity to dissect microbial communities into microscale volumes (μl, mg), containing hundreds to a few thousands of cells comprising picogram quantities of DNA. Such protocols will also aid efforts to characterize microbial communities in very low biomass environments. Currently, the most widely used commercially available low input DNA kit is Nextera XT (Illumina Inc., San Diego, CA, USA), which uses transposase insertion to fragment and tag the DNA with sequencing adapters. Using E. coli genomic DNA, a mock community of 54 phylogenetically diverse bacterial and archaeal isolates, and environmental samples obtained from coastal seawater, we evaluated a modified Nextera XT low input DNA library protocol for sequence yield and contamination, community composition fidelity, and assembly and population genome binning.

The manufacturer recommended Nextera XT library insert size range is 200 bp to 1 kb. We hypothesized that lowering the amount of input DNA may decrease average insert size due to a higher ratio of transposase to DNA resulting in more frequent tagmentations (cutting and attaching of adapters) per unit length of DNA. Indeed, a recent study by Bowers et al. (2015) found that insert size did indeed decrease as a function of input DNA for the Nextera XT kit. In theory, the insert size would only be restricted to a lower boundary of ∼20 nt due to steric interference between adjacent transposase complexes, which require at least a 19 bp binding site (Steiniger et al., 2006). The issue of over-tagmentation may be addressable by reducing reaction time or diluting the transposase. We observed a direct correlation between ATM dilutions (1:5 down to 1:50) and insert size, with the largest insert sizes at the highest dilutions (Fig. 1). It was surprising, however, that the 1 pg undiluted ATM controls produced larger than expected insert sizes, similar to the 1:10 dilutions (Fig. 1). A possible explanation is that the ATM includes a factor that inhibits the hyperactive Tn5 transposase and subsequently a dilution of the mix would also result in a dilution of the inhibitor, allowing an increase in enzyme efficiency. Inhibition of the Tn5 transposase activity is known for E. coli, in which an inhibitor of the transposition protein (a Tn5 transposase variant which lacks the N-terminal 55 amino acids and thus does not possess DNA-binding activity) forms a complex with Tn5 and interferes with transposition (de la Cruz et al., 1993). Based on these findings we prepared low input libraries using a 1:10 ATM dilution.

Average GC content is an often-reported metric in regard to Illumina libraries because of observed biases of GC-poor or GC-rich genomes and regions (Chen et al., 2013). We observed a slight decrease in the average read GC content with lower amounts of input DNA in the mock community libraries (Fig. 2). This was associated with a slight trend towards over-representation and under-representation of low and high GC organisms, respectively, with lower input DNA (Fig. S14). A slightly reduced coverage in high but also low GC regions was observed using the Nextera protocol (50 ng SOP) for virus genomes and the GC coverage trend was attributed to an amplification bias that occurred during the Nextera limited-cycle PCR (Marine et al., 2011). The opposite trend was observed by Bowers et al. (2015), which they attributed to organism specific differences. Another possible contributor to GC bias could be transposase insertion, as suggested previously (Lamble et al., 2013). An analysis of the regions immediately flanking the transposase insertion sites revealed a GC content ∼2% lower than the average read GC content, despite the known preference of Tn5 to insert at a guanine (Goryshin et al., 1998) (Fig. S15). However, this drop in GC content was consistent between the different input DNA concentrations suggesting that the limited cycle PCR is the main source of the observed slight shift in average GC content. This conclusion is in agreement with previous results who point to the PCR as the most important cause for GC-content bias during library preparation (Risso et al., 2011).

The largest difference in sequencing metrics between the Nextera XT 1 ng SOP and low input libraries, however, was the proportion of read duplicates, which increased dramatically with decreasing input DNA (Fig. 2). This is consistent with previous studies that found high levels of read duplicates for low input DNA libraries, whereby duplication levels of over 60% have been observed for 50 pg libraries at a sampling depth of five million reads, and up to 74% duplication occurring in 1 pg libraries with a sampling depth of 15 million reads (Chafee, Maignien & Simmons, 2015; Bowers et al., 2015). Read duplicate rates were positively correlated with ATM dilutions (Fig. 1), which is presumably the result of fewer transposase insertions producing fewer DNA fragments and subsequently less unique templates, increasing the likelihood of read duplicates during PCR. The number of limited cycle PCR cycles (12–20) had no consistent effect on the percentage of read duplicates (Fig. S10), thus at a given ATM dilution the amount of input DNA is the primary factor determining the fraction of read duplicates (Fig. S7). This is consistent with previous findings that PCR duplicates arise from a lack of DNA complexity (unique templates) due to low levels or quality of input DNA (Smith et al., 2014). While it is also possible that read duplicates are “community derived” and created by the fragmentation of two identical DNA molecules that are tagmented at exactly the same location, the probability of such an event in metagenomic samples is extremely low (Gomez-Alvarez, Teal & Schmidt, 2009). By subsampling reads we could show that read duplicates are library specific and their proportion is reduced by combining library replicates (Fig. S7B). This is likely due to the random loss of up to 90% of the initial DNA during library creation (Parkinson et al., 2012; Zhou et al., 2014), which in the case of Nextera XT libraries would be due to tagmentations that do not produce limited cycle PCR-amplifiable products. Thus, a repeat of this random process from the same starting material will result in reads covering different regions of the input DNA. Reducing read duplicates would appear to be a useful endeavor as they have been reported to bias coverage-based quality validation and hamper assemblies (Xu et al., 2012; Ekblom & Wolf, 2014). However, we found that the observed differences in percentage duplicates, GC content, and insert size (Fig. 2) had no impact on either our taxonomic or functional-based community profiles as evidenced by significant correlations between the 1 ng SOP and the low input libraries of the mock community (Figs. 3, S12 and S13). Furthermore, read duplicates per se did not affect assemblies (Fig. 4), which are instead limited by the number of unique reads in a sequence dataset (Fig. S16). Therefore, combining library replicates improves assembly outcomes because it increases the number of unique reads (Fig. S8). Based on these results, we suggest that low input DNA libraries are reproducible down to 100 fg using the minimally modified Nextera XT protocol of a 1:10 ATM dilution and 20 cycles of limited cycle PCR, regardless of the observed increase in duplicate reads. However, ≥ 1 pg libraries gave results more consistent with the SOP than the 100 fg libraries in terms of insert size, read duplicates, GC content, and community composition fidelity (Fig. 2). We therefore recommend 1 pg (∼102–103 cells) as the lowest DNA input amount using the modified protocol.

An important consideration when preparing low input DNA libraries is contamination. Small amounts of contaminating DNA are common in nucleic acid extraction kits and other laboratory reagents as indicated by no input DNA negative controls (Salter et al., 2014). Documented contaminants include representatives of the Proteobacteria, Actinobacteria, Firmicutes, Bacteroidetes, fungi, green plants and animals (Lusk, 2014; Strong et al., 2014). Such contamination is typically negligible in the context of standard input DNA libraries, but becomes an issue with decreasing input DNA, for example, with low biomass samples (Lusk, 2014; Salter et al., 2014). Therefore, we ran negative controls for both DNA extraction and library preparation in parallel with the low input DNA samples, where we substituted input DNA with ultrapure water. We recovered a near complete genome closely related to Methylobacterium aerolatum from the negative control libraries, which our data indicates can be mostly attributed to the UltraClean DNA extraction kit (Fig. 2). The genus Methylobacterium is a known contaminant introduced during sample preparation, possibly through molecular biology grade water, PCR reagents, or DNA extraction kits (Salter et al., 2014). One way to reduce potential reagent contamination is UV irradiation, which has been successfully used to decontaminate reagents for single-cell genomics (Woyke et al., 2011). However, since M. aerolatum, a strictly aerobic alphaproteobacterium, was originally isolated from air samples (Weon et al., 2008), aerial contamination cannot be entirely ruled out. The other major contaminant in the negative control libraries was human DNA, presumably attributable to the operator, though reagents may also be a source. Mining the remaining fraction of unmapped negative control reads, we found contaminants from a wide diversity of bacterial taxa (Fig. S11) matching previous reports (Lusk, 2014; Salter et al., 2014). Notably the family Staphylococcaceae was dominant in the library preparation negative controls with up to 100% relative abundance (Fig. S11). Since this family includes the genus Staphylococcus, a known member of the human skin and mucus microbiome (Otto, 2010), we attribute its presence to operator introduced contamination. We therefore recommend running ultrapure water controls with and without the DNA isolation procedure to identify reagent-specific contaminants, and to include them in contaminant screens of low input DNA sequencing libraries.

Having addressed contamination issues and established the preservation of community composition for the low input DNA libraries in the mock community, we applied the modified Nextera XT protocol to marine samples, one of the most intensively studied ecosystems using metagenomics (Gilbert & Dupont, 2011; Reddy et al., 2015). We extracted DNA from samples having a range of volumes, representing standard (10 L filtered control) down to microscale (10 μl) samples. We also prepared libraries from dilutions of the 10 L control to match the DNA inputs of the low volume samples. Ten microliter samples were the lowest volumes used as they were estimated to comprise ∼3 pg of DNA, which is above our recommended 1 pg library input DNA threshold (Fig. 5). Community profiles based on 16S rRNA sequences identified in the samples were consistent with those of previously described marine surface waters (Fig. 6). As was observed with the mock community, both the taxonomic and functional profiles of the low volume and filtered dilution control samples were strongly correlated to the 10 L filtered SOP libraries when comparing replicate averages (Figs. 7, S19 and S20). However, we noticed a high degree of variance between replicates of the 10 μl libraries (Figs. 8 and S18), above the expected technical variation observed in the corresponding 5 pg filtered dilution replicates of the taxonomic profiles (Fig. 9). We attribute this elevated variation to microscale biological differences between replicates of the same sample volume. Indications of such microscale patchiness have been reported previously for bacterioplankton with significant variation in bacterial species richness amongst microliter samples using denaturing gradient gel electrophoresis (Long & Azam, 2001). Therefore, our results constitute the first metagenomic data of microscale heterogeneity in marine surface waters, which has mostly been overlooked to date (Azam & Malfatti, 2007; Stocker & Seymour, 2012). Indeed, the observed microheterogeneity may be an underestimate of in-situ conditions as the low volume samples were taken from a 10 L subsample in which localized mixing may have occurred.

De novo assembly and binning are important bioinformatic steps in metagenomic workflows (Leung et al., 2013), which we assessed in relation to low input DNA libraries using the mock community and marine datasets. It is important to consider that 1 pg of DNA only comprises ∼1.2 Gb of potentially sequenceable template (cf. ∼1.2 Tb template for the 1 ng SOP), therefore expectations of assembly and binning should be appropriately calibrated. Both steps are known to be dependent on community composition with the general rule of thumb that increasingly complex communities produce lower quality assembly and binning results (Mavromatis et al., 2007; Dröge & McHardy, 2012). Therefore, we did not directly compare the mock to marine results, instead focusing on within community type comparisons. As expected, both assembly and binning deteriorated with decreasing input DNA, although substantial assemblies (kb range) were still obtained down to 1 pg libraries for both community types (Figs. 4 and S21), which are suitable for e.g. gene neighborhood analyses. Approximately a third of the population bins (> 50% completeness) obtained for the mock community SOP were still recoverable from the 1 pg datasets (Table 1), in contrast to a recent report using a simpler mock community in which no bins of this quality were recovered from 1 pg Nextera XT libraries (Bowers et al., 2015). The low volume marine datasets yielded no bins, although only five bins were obtained from the marine SOP in total (Table S5), consistent with the higher complexity of this community, again emphasizing the case-by-case nature of assembly and binning.

Conclusions

We demonstrate that it is possible to successfully prepare and sequence low input metagenome libraries down to 100 fg of DNA using a slightly modified version of the Nextera XT protocol, with the important caveat that negative controls are included to detect possible reagent and other contaminants. Community composition was highly reproducible down to 1 pg despite a pronounced increase in the proportion of read duplicates with decreasing input DNA indicating that duplicate formation is random. Assembly and binning are both compromised by lowering input DNA due to decreasing sequenceable template and associated number of unique reads. However, both assembly and binning are still possible in the pg range depending on community complexity. Applying the approach to surface marine waters, we found evidence for microscale community composition heterogeneity in 10 μl volumes, demonstrating the utility of applying metagenomics at spatial scales relevant to microorganisms.

Supplemental Information

Supplemental Information 1 Supplementary Tables and Figures.

Click here for additional data file.

We thank Mircea Podar and Stuart Denman for generously providing DNA for construction of the mock community, David Wood and Pierre-Alain Chaumeil for providing bioinformatics support, Yun Kit Yeoh for contributing reference population genomes bins, Paul Evans for assistance with the mock community, and the ACE team (http://ecogenomic.org) for stimulating discussions.

Additional Information and Declarations

Competing Interests

Author Contributions

Field Study Permissions

DNA Deposition

Data Deposition

The authors declare that they have no competing interests.

Christian Rinke conceived and designed the experiments, performed the experiments, analyzed the data, contributed reagents/materials/analysis tools, wrote the paper, prepared figures and/or tables, reviewed drafts of the paper.

Serene Low performed the experiments, contributed reagents/materials/analysis tools, reviewed drafts of the paper.

Benjamin J. Woodcroft analyzed the data, contributed reagents/materials/analysis tools, reviewed drafts of the paper.

Jean-Baptiste Raina contributed reagents/materials/analysis tools, reviewed drafts of the paper.

Adam Skarshewski analyzed the data, reviewed drafts of the paper.

Xuyen H. Le analyzed the data, reviewed drafts of the paper.

Margaret K. Butler contributed reagents/materials/analysis tools, reviewed drafts of the paper.

Roman Stocker reviewed drafts of the paper.

Justin Seymour reviewed drafts of the paper.

Gene W. Tyson conceived and designed the experiments, reviewed drafts of the paper.

Philip Hugenholtz conceived and designed the experiments, wrote the paper, reviewed drafts of the paper.

The following information was supplied relating to field study approvals (i.e., approving body and any reference numbers):

There is no permit required in Australia to sample seawater from the Ocean.

The following information was supplied regarding the deposition of DNA sequences:

We generated a mock community of 54 Bacteria and Archaea (Table S1). Ten mock community genomes were sequenced and assembled in this study, the sequences were deposited in NCBI-BioProject under the BioProject ID PRJNA324744.

The following information was supplied regarding data availability:

The adapter trimmed fastq read files were deposited on the Microscale Ocean webpage (microscaleocean.org/data/category/9-low-input-dna-libraries-peerj).

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
