# Peer review of "Validation of picogram- and femtogram-input DNA libraries for microscale metagenomics"

_PeerJ, doi:10.7717/peerj.2486_

## Round 0.1 · original submission · Minor Revisions

As you can see from the detailed reviewer reports, both expert reviewers consider your work as a substantial addition to the field of environmental metagenomics, as it allows to further reduce the amount of input DNA needed for relatively unbiased assessment of the microbial coding capacity.

I am very much looking forward to a further improved revised manuscript, that should consider the comments of the reviewers regarding, among others, further streamlining with respect to key figures and supplementary material, and adding further technical detail and some basic biological interpretation.

Reviewer 1 ·

Basic reporting

Performing metagenomic sequencing on minute environmental DNA samples continues to be a challenge. While various DNA amplification techniques can provide sufficient DNA, most approaches result in introduction of biases, which are problematic when quantitative ecological analyses are among the goals. Here, Rinke et al used samples with defined composition “mock communities” to test experimental and informatics procedures to push the limit of DNA amounts in metagenomic experiments down several orders of magnitude below the standard approaches. They successfully demonstrated that with minimal changes in the protocols one can in fact reconstitute the community structure using the DNA equivalent of hundreds of cells and applied it to actual environmental samples.

Experimental design

DNAs were diluted to represent the equivalent of small populations of cells. Overall I have no issues with the ways they did this, although I am surprised they got DNA eluted from kits that are designed to work with relatively large quantities of material. Clearly though contamination from kit reagents remains a problem. There is no mention of attempts to irradiate with UV all the reagents and the kit component, I suspect that may have reduced that contamination? If that was not done perhaps they should mention it as a future possible way to address the issue.

Validity of the findings

An improved approach to access the metagenome of minute samples and communities with very low biomass.

Reviewer 2 ·

Basic reporting

See "General Comments for the Author"

Experimental design

See "General Comments for the Author"

Validity of the findings

See "General Comments for the Author"

Additional comments

In this manuscript, Rinke et al. tested the lower limit of metagenomic library preparation using Illumina’s Nextera XT low biomass library preparation kit by extending its minimum starting material by approximately 10,000 fold (Illumina recommends 1 ng starting material), going down to 100 fg starting material. The authors analyzed resulting libraries of various input material that were derived from a 54 member mock community and a marine surface water sample.

This study extends the existing literature on low template DNA metagenomic library preparation and sequencing by (i) analyzing mock and environmental communities down to 100 fg starting material, (ii) testing the efficacy of these low biomass preps on an environmental sample where the control was a large volume (10L) water sample with a high conc of input cells (~10^7-10^8), and (iii) not only testing these protocols on diluted DNA, but on very low volume marine samples, ~ 10ul where there only around 10^3 starting cells per 10ul sample which can be taken advantage in studies that require small area/volume samples due to microenvironment heterogeneity.

The study is thorough and should be of great interest to the Peer J readership.

GENERAL COMMENTS:

(1) I very much like the microscale angle of the paper, but can the authors explain how the actually maintained biologically meaningful communities using surface water samples? “Low volume samples (1 ml, 100 μl, 10 μl) were pipetted individually from unfiltered bulk seawater and snap frozen.” How do you prevent complete mixing of the unfiltered bulk seawater? A few explanatory words would be useful for the reader.

(2) While I see the rationale for using surface marine waters and this is indeed the first metagenomic data of microscale heterogeneity in marine surface waters, I was disappointed not to read more about any biological insights, even just at a high level. What did the authors learn? Any biological meaningful and impactful conclusions that could be drawn from these data?

(3) The manuscript is nicely detailed and many analyses were performed. For the sake of the reader, I wonder if it could be streamlined or at least some of the Figures could be moved to the supplement. Some Figures seem less critical than others. Key Figures in my opinion include Figures 1, 2, 4, 5, 8, etc. Could 3, 6, 7 and possibly others be moved to the supplement to emphasize the main message? Anyone interested in more details, can go to the supplement.


MINOR, MORE SPECIFIC COMMENTS:

Abstract:
- Instead of “limiting template DNA” maybe “limited DNA complexity”
- “The main technical issues with the low input libraries were a greater potential for contamination, limiting template DNA which compromises assembly and binning, and an associated higher percentage of read duplicates.” Maybe reword to “.. which has a direct effect on assembly and binning..”.

Line 63 ff:
The authors portrait the common workflow as being non-tagmentation based library prep (“usually follows a common workflow.. ”). I would disagree with this. Most labs have meanwhile adopted nextera tagmentation-based library prep and are getting away from mechanical fragmentation and ligation-based adapter addition (chiefly for cost and simplicity; no corvaris etc needed).. suggest rewording. The method can be presented as one of the methods, but not the most common one.

Line 81f:
Period missing after sentence. One may also include the fact that MDA reagents can be contaminated (ie Delftia acidovorans) and that random hexamer primed whole genome amplification may bring up such contaminants during amplification, greatly skewing the results.

Lines 103ff:
This section is a bit confusing. 10 of the 54 reference genomes used in this analysis were obtained through genome binning (i.e. population genomes). This part of the statement is fine, however to make the mock community, where was the genomic DNA from these 10 bins obtained from? Along the same lines, much of the mock community DNA was obtained from collaborators, but it might good to list those collaborators or labs from which the mock members were obtained in Table S1.

Lines 130ff:
I may have missed it, but is there a specification on how the DNA extracted from the water sample was quantified? The authors used /4 of the DNA extract (5ul)’ but how much DNA does this equate to? 1pg for all samples by chance and no details on quantification? I suspect the DNA concentartion is too low to be quantified even with picogreen, but can the authors at least say something about it? You extrapolate from the numbers of cells in the given volume? Did you do cell counts using flow cytometry? Additional details here would be helpful.

Line 214:
“reverse reads were subsampled to a depth of 5 million”. Suggest to reword to “reverse reads were subsampled 5 million reads”.. It’s not really a depth, rather than just the number of reads.

Line 215:
“and then subsampled to a depth of 25 million”. Likewise, I’d suggest to reword to “and then subsampled to 25 million reads”.

Figure 1:
- Can the authors make modify the names for the samples in the Figures (ie: “Ecoli_1pg_1_10” could read more clearly “ E. coli 1pg (1:10 dilution)”
- Is there no data on the E. coli 1ng SOP (panel a)?

Figure 2:
- What’s the “new” in “1pg-new-rep1” etc..? Again, the names could be modified to make them more accessible to the reader (this is not a lab presentation, but a publication).
- Could the authors identify what the remaining “unmapped” sequence could be by protein blast or marker gene phylogenetic analysis? It would be in interesting to know, as not an insignificant amount of data.

Line 367ff:
“Therefore, we suggest that community composition can be reliably assessed using low input libraries down to 100fg despite higher proportions of read duplicates (Fig. 3a)”. But what about the higher amount of contamination? Looking at Figure 2, I would disagree with the author’s statement. 100 fg libraries can lead to significant amount of contaminant sequences (here mostly human and un-mapped). While I agree that the community composition correlation looks OK for 100 fg libraries (although the highly contaminated replicate was omitted in the analysis, which seems inappropriate!), based on the contaminant results, my recommendation would be 1pg, not 100 fg. Human probably won’t assemble well, but if you have some contaminants with a smaller genome, you can get significant skew in your results.

Figure 9:
Not publication-ready. Tax strings need to be cleaned up.

Line 591ff:
“Approximately a third of the population bins (>50% completeness) obtained for the mock community SOP were still recoverable from the 1pg datasets (Table 1), in contrast to a recent report using a simpler mock community (Bowers et al., 2015).”
Can you say more here? I believe in Bowers et al, more than half of the population bins (>50% completeness) that were obtained for the mock community SOP were still recoverable from the 5pg dataset. I think that would be worth pointing out!

Figure 5:
Spelling error: Figure 5 text box second sentence should read, ‘red bars should indicate’, not ‘read bars indicate’.

Figure 9:
Place text on legend in Fig. 9, otherwise not sure what 0- ~800 means

Figure S1:
Figure shows that functional profile is performed before assembly. This is probably fine, but won’t functional profiling work better on assembled contigs (larger sequences = better functional assignments and gene linkages) as opposed to reads? Although maybe they were comparing KEGG composition across libraries, which might be more straightforward when only using reads on a subsampled library (i.e. 5 million reads across all libraries).

Fig S12B.
May be more readable if the authors only include panel A?

---

## Round 0.2 · accepted · Accept

All concerns raised by the reviewers have been addressed appropriately.

Reviewer 2 ·

Basic reporting

See general comments

Experimental design

See general comments

Validity of the findings

See general comments

Additional comments

The authors addressed all of the concerns in their revision and the study should be published. Nice work.